# Research Advances on the Stability of mRNA Vaccines

**DOI:** 10.3390/v15030668

**Published:** 2023-03-02

**Authors:** Feiran Cheng, Yiping Wang, Yu Bai, Zhenglun Liang, Qunying Mao, Dong Liu, Xing Wu, Miao Xu

**Affiliations:** 1National Medical Products Administration Key Laboratory for Quality Research and Evaluation of Biological Products, Institute of Biological Products, National Institutes for Food and Drug Control, Beijing 102600, China; 2National Health Commission Key Laboratory of Research on Quality and Standardization of Biotech Products, National Institutes for Food and Drug Control, Beijing 102600, China; 3Center for Reference Materials and Standardization, National Institutes for Food and Drug Control, Beijing 102600, China

**Keywords:** mRNA, vaccines, stability, lipid nanoparticle, degradation

## Abstract

Compared to other vaccines, the inherent properties of messenger RNA (mRNA) vaccines and their interaction with lipid nanoparticles make them considerably unstable throughout their life cycles, impacting their effectiveness and global accessibility. It is imperative to improve mRNA vaccine stability and investigate the factors influencing stability. Since mRNA structure, excipients, lipid nanoparticle (LNP) delivery systems, and manufacturing processes are the primary factors affecting mRNA vaccine stability, optimizing mRNA structure and screening excipients can effectively improve mRNA vaccine stability. Moreover, improving manufacturing processes could also prepare thermally stable mRNA vaccines with safety and efficacy. Here, we review the regulatory guidance associated with mRNA vaccine stability, summarize key factors affecting mRNA vaccine stability, and propose a possible research path to improve mRNA vaccine stability.

## 1. Introduction

Vaccine stability refers to the ability of a vaccine to maintain its physical, chemical, microbiological, and biological properties during its shelf time. To ensure vaccine quality, it is necessary to study and monitor the stability of the vaccine over its full life cycle, including the bulk, the final product, and the period of their transportation and storage until the end of its shelf life. Messenger RNA (mRNA) serves as a template for protein translation [1]. The coronavirus disease 2019 (COVID-19) pandemic has accelerated the development of mRNA vaccines. According to the World Health Organization (WHO), two COVID-19 mRNA vaccines have been approved and over forty have entered clinical trials. Currently, there are also mRNA vaccines entered in clinical trials for the prevention and control of infectious diseases, cancer and genetic diseases. Due to the instability of mRNA, the vaccine produced by Pfizer needs to be kept below −70 degrees Celsius, while Moderna’s vaccine only needs to be kept below −20 degrees Celsius, which has restricted access to effective immunization for populations in underdeveloped countries and resulted in the discarding of a large number of prepared vaccines [2]. Thus, stability is a bottleneck for the development and application of mRNA vaccines.

The stability of mRNA vaccines has been studied both in vivo and in vitro. In vivo stability is associated primarily with the inherent properties of mRNA, including the optimization of the regulatory region, coding sequence, 5′-cap, poly-A tail, and untranslated region of the mRNA. In vitro stability is associated primarily with critical quality attributes (CQAs), such as mRNA integrity, fragment length, lipid nanoparticle (LNP) particle size, and lipid composition. Factors including mRNA structure, LNP delivery systems, excipients, and manufacturing processes could affect mRNA vaccine stability. Given the inherently unstable characteristics of mRNA vaccine components, improving stability is a key method to enhance the effectiveness of mRNA vaccines.

To enhance and guide vaccine development and quality control, the WHO, International Council for Harmonization of Technical Requirements for Pharmaceuticals for Human Use (ICH), and various national drug regulatory authorities have issued regulations and guidelines for stability studies and introduced a new concept of risk-based stability prediction and statistical models for vaccine stability studies in recent years [3]. More importantly, exploring the mRNA vaccine degradation mechanisms and influencing factors could contribute to mRNA vaccine development in the future.

## 2. Regulatory Documents and Guidelines Related to Vaccine Stability

The WHO published the document Stability of Vaccines in 1989 [4]. A 1998 revision emphasized the effect of temperature on vaccine stability [5]. The 1995 document, ICH Q5C Quality of Biotechnological Products: Stability Testing of Biotechnological/Biological Products, mentioned that studies on biological product stability are based on appropriate testing of biological activity [6]. In 2006, the WHO guidelines for stability evaluation of vaccines suggested that stability study parameters should be selected based on the characteristics of different vaccines [7]. The 2011 revision of these guidelines suggested that stability studies should examine environmental factors, such as temperature, light, freeze-thaw, and physical stress [8]. The United States Pharmacopeia, European Pharmacopoeia, Chinese Pharmacopoeia, and relevant regulatory authorities such as the United States Food and Drug Administration (FDA), European Medicines Agency (EMA), and Center for Drug Evaluation (CDE) of the National Medical Products Administration of the People’s Republic of China have also proposed requirements for vaccine stability studies [9,10].

For mRNA vaccines, the European Union issued the Investigational Medicinal Product Dossier document in 2006. The document stated that the CQAs of mRNA vaccine products include mRNA integrity, content, and potency [11]. In 2020, the CDE issued the Technical Guidelines on Research and Development of COVID-19 Prophylactic mRNA Vaccines (draft edition). These guidelines emphasized that mRNA vaccine stability studies and evaluations should follow the relevant guidelines for stability studies of biological products. In addition, conditions to be examined should include temperature, pH, light, humidity, and a number of freeze-thaw cycles. Finally, there should be a focus on physicochemical properties and expression efficiency, such as encapsulation efficiency, particle size, and active ingredient content [12]. In 2021, the WHO issued a guideline document stating that mRNA vaccine stability indicators include appearance, mRNA content, vaccine potency, mRNA integrity, encapsulation efficiency, particle size, polydispersity, and impurities associated with mRNA and lipids [13]. In addition, accelerated stability and stress studies must be carried out to support shelf-life prediction.

An mRNA vaccine contains lipid nanoparticles and nucleic acids. The Liposomal Drug Products: Chemistry, Manufacturing, and Controls; Human Pharmacokinetic and Bioavailability; and Labeling Documentation document issued in 2018 by the FDA noted CQAs for liposome drug products, including particle size, size distribution, and morphology, and emphasized studying the physical, chemical, and microbiological stability of liposomal drugs, as well as the stability of lipid components in liposomal drugs and enclosed drugs [14]. In 2022, the CDE published the Technical Guidelines for the Pharmacological Study and Evaluation of In Vivo Gene Therapy Products (draft edition). The document lists the corresponding technical requirements for in vivo gene therapy products, such as nucleic acid-based products, and states that product design should consider elements such as the 5′-cap, poly-A tail, nucleic acid modifications, and delivery systems, which also affect mRNA stability [15].

The stability guidelines and regulations issued by the WHO and agencies in the U.S., Europe, and China all include results, statistics, and analysis of stability studies. Among them, the ICH Q1E is a comprehensive and specific guideline for the statistical analysis of stability data [16]. In addition, the document proposes a general framework for time-series regression analysis of quantitative indicators (e.g., content, degradation products) and conditions for combining different batches for shelf-life prediction. Analytical software predicts the expiration date by determining the earliest time at which the 95% confidence limits of the mean values intersect with the proposed acceptable criteria. Statistical analysis of the stability results enables the identification of fluctuations in product stability and provides a scientific basis for setting the validity period (Table 1).

## 3. Advances in mRNA Vaccine Development

The mRNA vaccine is a new type of vaccine that contains exogenous target gene sequences prepared through in vitro transcription and synthesis. The delivery of these sequences into somatic cells stimulates a specific immune response and confers immune protection by using the somatic cells to express protein. In recent years, key advances in mRNA modification and delivery systems have improved the translation efficiency, stability, immunogenicity, and safety of mRNA. Compared to conventional vaccines, mRNA vaccines have three advantages. The first is safety. mRNA and its delivery systems can be degraded through normal cellular metabolism and are not integrated into the genome of the somatic cells. The production process does not involve live viruses, and the biosafety risk is low. The second advantage is the short development cycle time. After sequence design and validation, mRNA can be transcribed directly in vitro. The third advantage is the high production capacity. With the ability to rapidly scale up production to billions of doses, the vaccination needs of new outbreaks of infectious diseases could be easily fulfilled.

In 1989, Malone et al. [20] proposed the concept mRNA as a therapeutic drug and successfully transfected a variety of eukaryotic cells using mRNA encapsulated in liposomes. A 1990 paper by Wolff et al. [21] described the injection of naked mRNA synthesized in vitro directly into mouse skeletal muscle cells. The authors demonstrated that exogenous mRNA resulted in direct expression of the encoded protein in animal tissues. In 2005, Karikó et al. [22] reported that mRNA modification reduced the level of innate immune response and decreased the release of inflammatory factors. On 23 August 2021, BNT162b2 (Comirnaty^®^, New York, NY, USA) was approved by the FDA as the first officially approved COVID-19 vaccine globally. To date, there are 41 COVID-19 mRNA vaccines in clinical trials worldwide [23] (Table 2).

mRNA vaccines are currently undergoing clinical trials for the prevention and control of diseases other than COVID-19 vaccines, including vaccines for influenza [24], human immunodeficiency virus (NCT05001373), Zika virus (NCT04064905), respiratory syncytial virus (NCT04528719), and cytomegalovirus (NCT04232280). In cancer research, tumor vaccines are being developed for melanoma (NCT03897881), breast cancer (NCT02316457), ovarian cancer (NCT04163094), non-small cell lung cancer, and colorectal cancer (NCT03948763). In the treatment of genetic diseases, clinical trials are ongoing for mRNA vaccines for cystic fibrosis [25], propionic acidemia [26], and transthyretin amyloidosis [27].

In addition to linear mRNA, circular RNA and self-amplifying RNA (saRNA) technologies are rapidly progressing. The unique structure of circular RNA makes it more stable than linear RNA and suggests the potential for vaccine development. saRNA encodes RNA polymerase and uses its own RNA sequence as a template for replication, achieving protein expression levels comparable to those of conventional mRNA but at lower doses.

## 4. Mechanism of mRNA Vaccine Degradation

### 4.1. Degradation of mRNA

The mRNA molecule is naturally both physically and chemically unstable. The physical instability of mRNA includes the loss of secondary or tertiary structure, as well as aggregation and precipitation, which affect the translation of mRNA molecules [28]. Compared to physical instability, chemical degradation has a greater impact on the stability of nucleic acids [29]. Chemical degradation includes hydrolysis and oxidation. Similarly, mRNA hydrolysis is affected by base sequence [30] and mRNA secondary structure [31,32]. mRNA molecules are degraded by protein-based nucleases, ribozymes, and acids and bases through similar mechanisms.

Oxidation can lead to base breakage, strand breakage, and changes in mRNA secondary structure [29]. However, hydrolysis is still considered the main factor leading to mRNA degradation. In addition, the degradation pattern of RNA molecules is consistent with the Arrhenius model [33].

### 4.2. Degradation of LNPs

In addition to mRNA as an active substance, components of LNP can interact with each other, thus affecting the stability of the product. LNP instability primarily manifests as physical and chemical instability. Physical instability can be categorized as aggregation, fusion, or content leakage. During storage, LNPs are prone to aggregation and fusion [34,35]. Polyethylene glycol (PEG) can prevent LNP aggregation and thus improve LNP stability [36]. In addition, physical degradation can also lead to leakage of mRNA, which affects the stability of the encapsulated product. Under normal conditions, the encapsulation efficiency of mRNA typically exceeds 90%. Naked mRNA cannot easily enter cells and is prone to degradation, resulting in translation failure. Chemical instability is primarily due to hydrolysis and oxidation of LNP lipids. For example, the carboxylic ester bonds in lipids, such as DSPC and the ionizable cationic lipids, are highly susceptible to temperature- and pH-dependent hydrolysis during storage. Impurities generated from LNP lipid oxidation are highly susceptible to interaction with nucleic acid molecules, which can lead to oxidation of encapsulated mRNA. For example, aldehydes can react with DNA bases [37,38] and mRNA [39]. Packer et al. found that ionizable lipids in the LNP component can produce aldehyde impurities through oxidation and hydrolysis pathways and can form covalent adducts with mRNA, thus affecting mRNA biological activity [39]. PEG lipids are an important potential source of hydroperoxides, which are often unstable and whose degradation can lead to chain reactions and catalyze the degradation of ionizable lipids. Lipid degradation products can also react with mRNA, leading to the formation of covalent mRNA-lipid adducts.

Packer et al. [39] constructed a model for mRNA vaccine stability prediction by storing mRNA vaccines at −20, 5, 25, or 40 °C and using reversed-phase ion-pair high performance liquid chromatography (RP-IP HPLC) to detect the formation of adducts between mRNA molecules and lipid degradation products, which directly affect the level of mRNA translation and vaccine activity. Based on three consecutive months of assay data, the authors developed a kinetic model for mRNA vaccine expiration dates and used the model to predict expiration dates of mRNA vaccines. In 2021, a review on ICH Q1A(R2) and Q5C by McMahon et al. [3] stated that the use of stability models and risk-based predictive stability can guide the assignment of retest periods and shelf life for new products under long-term storage. According to an EMA document, a stability study of the Moderna COVID-19 vaccine mRNA-1273 found that the degradation pattern of the mRNA molecule also demonstrated Arrhenius behavior with first order kinetics. In addition, the stability profiles in Moderna’s vaccine stability study were predictable and amendable to modeling, enabling a good understanding of the chemical degradation process.

## 5. Factors Affecting mRNA Vaccine Stability

mRNA vaccine stability is closely associated with mRNA structure, excipients, and the LNP delivery system. In addition, the manufacturing process plays a key role in stability. This section focuses on the factors influencing mRNA vaccine stability.

### 5.1. mRNA Structure

#### 5.1.1. Integrity and Length

The integrity of the mRNA molecule, including the 5′ cap, coding region, untranslated region (UTR) and poly A tail, is a CQA that affects mRNA vaccine stability and efficacy. The deletion of mRNA fragments, such as stop codon deletion, has been shown to significantly shorten the half-life of mRNA, leading to degradation and translation failure [40].

mRNA fragment length also affects mRNA stability. A negative correlation has been reported between mRNA length and half-life in yeast [41]. Under the same storage conditions, a significant decrease in half-life occurred with increasing RNA length [31]. Guillaume et al. [42] reported that unlike the first-generation mRNA-1273 vaccine encoding the full-length sequence of the severe acute respiratory syndrome coronavirus 2 (SARS-CoV-2) S protein, the second-generation Moderna COVID-19 mRNA vaccine mRNA-1283 encoding the shorter N-terminal and receptor-binding domain of the S protein had significantly improved mRNA stability, with an extension in shelf life from 6 to 12 months at 2–8 °C.

Onpattro^®^ is an approved small interfering RNA (siRNA)-based gene product that uses an LNP delivery system with components similar to those of the two approved COVID-19 mRNA vaccines. It has a shelf life of 3 years at 2–8 °C with no significant changes in the indicators during storage [35,43]. These findings also imply that the inherent stability of mRNA is a key factor in determining the shelf life of mRNA-LNP vaccines.

#### 5.1.2. Sequence Optimization and Modification

Sequence optimization and mRNA modification could be the major focus for improving mRNA-LNP vaccine stability. Mature eukaryotic mRNAs include a 5′-cap, a coding region, UTR, and a 3′ terminal poly A tail.

mRNA optimization includes codon optimization in the coding sequence as well as UTR optimization. Replacing rare codons with common codons increases mRNA translation [44]. Increasing GC content also enhances mRNA translation levels [45,46,47] and decreases immunogenicity [48]. Concerning the optimization of mRNA UTRs, the introduction of the Kozak consensus sequence after the 5′ UTR can also improve the stability and translation initiation level [49]. The selection of human α- and β-globin 3′ UTR sequences significantly improves mRNA stability and translation efficiency [50,51]. The introduction of the 3′ UTR sequences of two β-globins in a head-to-tail manner between the coding sequence and the poly A tail further improved mRNA stability and translation efficiency [52].

The 5′-cap is a protective structure resulting from a structural modification on the first 7-methylguanosine at the 5′ end of mRNA linked to the first nucleotide via a triphosphate linkage (Cap0, m7GpppN). Cap0 inhibits the degradation of mRNA by nucleases through spatial binding with transcription initiation factors, and increases translation initiation levels. Continued methylation of Cap0 to form Cap1 and Cap2 further reduces the level of immunogenicity.

The poly A tail usually consists of 10–250 nucleotides and improves mRNA stability, resistance to nuclease degradation and translation by forming a complex with poly A binding protein [53]. In addition, the length of the poly A tail affects translation efficiency and protein expression levels [52,54]. Typically, the poly A tail of mRNA is composed of adenosine chain. However, a recent study reported that mRNA tails containing cytosine (C) improve the level and duration of mRNA expression [55]. In addition, cytosine (C) substitution can improve resistance to mRNA degradation, thus prolonging its half-life.

The various nucleotide modifications of mRNA include pseudouridine substitution and uracil and cytosine methylation. Sequence modifications reduce the inflammation-stimulating effects of mRNA and cleavage by nucleases, improve mRNA stability, and enhance translation levels [22,56]. The two approved COVID-19 mRNA vaccines, mRNA-1273 (Moderna) and BNT162b2 (Pfizer/BioNTech), were both designed with N1-methyl pseudouridine triphosphate (m1ψTP) instead of uridine triphosphate (UTP). m1ψTP has a better safety profile and more effectively reduces immune stimulation compared to pseudouridine [57]. These findings also suggest that the introduction of m1ψTP in mRNA can effectively reduce the immunogenicity of mRNA vaccines and enhance mRNA stability and protein expression.

In addition, sequence optimization and modification, such as increasing GC content and introducing pseudouridine, could enhance the stability and translation level of mRNA by stabilizing its secondary structure and increasing base stacking and melting point [58,59,60,61].

#### 5.1.3. Circular RNA and Self-Amplifying RNA Structures

Circular RNAs (circRNAs) are single-stranded, closed-loop RNAs that are covalently bonded without a 5′ cap or 3′ tail structure. circRNA can potentially encode protein and serve as a foundation for vaccine development. The unique circular structure of circRNA makes it more resistant to nuclease degradation and results in a longer half-life than linear RNA [62,63,64,65]. Moreover, circRNA does not contain stop codons and is able to reduce the frequency of ribosome dissociation from RNA, which in turn increases protein translation and expression levels. circRNA can be expressed consistently in cells for 5 to 7 days, whereas linear mRNA with the same sequence is reportedly only consistently expressed for 2 days [66,67]. The stability of a circRNA vaccine developed by Qu et al. [68] using LNPs was also significantly better than that of linear mRNA vaccines at 4, 25, and 37 °C.

In addition to expressing the target protein, self-amplifying RNA vaccines also carry a sequence for RNA-dependent RNA polymerase. The expressed RNA polymerase uses its own RNA sequence as a template for self-amplification to produce more copies of saRNA. The major advantage of saRNA is that it can achieve protein expression levels similar to those of conventional mRNA at low doses [69,70]. Moreover, since saRNA produces double-stranded RNA during transient replication, this RNA also increases the level of innate immune activation in the body.

### 5.2. Excipients

As stated previously, key active pharmaceutical ingredients for mRNA vaccines include cap analogs, ribonucleoside triphosphates (NTPs), modified nucleotides (m1ψTP), and polynucleotide tails. These play an important role in improving mRNA stability, reducing immune stimulation, and enhancing mRNA expression and safety.

The excipients in mRNA vaccine formulations include LNP lipids, sucrose, and the buffer for injection. Lipids are major components of vaccines and play an important role in mRNA delivery. mRNA vaccines require cryopreservation, and sucrose is often added as a protective agent to ensure vaccine freezing. In addition, the buffer affects the long-term mRNA vaccine stability.

Since LNP lipids are susceptible to oxidative hydrolysis during storage, the selection of high-quality lipid excipients is essential to prevent impurities that affect lipid stability, such as hydroperoxides and metals [71].

Ribozymes can be catalytically active even at −20 °C [72], and some remain active even at −70 °C [73]. For this reason, the excipients used in mRNA vaccine formulations must be free of RNAase.

Sucrose is a common excipient used to reduce the crystallization temperature and ice crystal formation in the aqueous phase. The interfacial film formed by the adsorption of sugar molecules at the oil-water interface also increases the steric hindrance and electrostatic repulsion between droplets. Ayat et al. [34] showed that the addition of sucrose increases the freeze-thaw stability of mRNA vaccines. Sucrose was also selected as a vaccine stabilizing agent and cryoprotectant in the Moderna and Pfizer/BioNTech COVID-19 mRNA vaccine formulations. It ensures the lipids do not become too sticky during cold storage. Recently, Mao et al. [74] reported the development of a needle-free injectable COVID-19 mRNA vaccine. This formulation utilizes sucrose as an antioxidant for LNP particles. The purported stability of this vaccine is 6 months at 2–8 °C.

### 5.3. LNP Delivery Systems

mRNA delivery vehicles primarily include LNPs, polymeric nanoparticle delivery systems, inorganic nanoparticle delivery systems, and peptide nanoparticle delivery systems. Among them, LNPs are the most widely used in mRNA vaccines; they are also common non-viral delivery vehicles in gene therapy [75]. mRNAs can be protected by LNPs from ribonuclease degradation and enter cells by endocytosis then escape into the cytoplasm via endosomes to complete mRNA translation and expression [76]. The immunogenicity of mRNA vaccines is also increased after LNP encapsulation and delivery [77].

LNPs in mRNA vaccines are composed of four main components: phospholipids, cholesterol, PEG lipids, and ionizable lipids. Because phospholipids can spontaneously organize into lipid bilayers and have high phase transition temperatures, they are often used as structural lipids and account for approximately 10–20% of the total lipids in LNPs. Kauffman et al. [78] found that phospholipid 1,2-dioleoyl-sn-glycero-3-phosphoethanolamine (DOPE) improved mRNA transfection efficiency. Zhang et al. [79] found that DOPE can affect the biodistribution of LNPs in vivo. Cholesterol is another structural lipid in LNPs, accounting for approximately 20–50% of the total lipids in LNPs. Cholesterol enhances particle stability by modulating membrane integrity and rigidity [80,81]. The molecular geometry of cholesterol derivatives can further affect the delivery efficiency and biodistribution of LNPs [82]. PEG lipids serve to control the size of LNPs and stabilize them, preventing aggregation of the particles during storage [83,84]. PEG lipids can also prevent rapid uptake of LNPs, reducing serum protein-mediated opsonization and clearance by reticuloendothelial cells and improving the circulation half-life of LNPs [85]. In addition, functional modifications of PEG lipids can facilitate the binding of LNPs to ligands or biomolecules [86]. Ionizable lipids account for 30–50% of the total lipids of LNPs. Ionizable lipids contain tertiaryamine groups and are only neutral at a physiological pH when present in the core of the LNPs, but become positively charged when ionized at a low pH in acidic endosomes. Since neutral lipids interact less with the anionic membrane of vascular cells and greatly improve the biocompatibility of LNPs, pH-sensitive ionizable lipids are more favorable for in vivo mRNA delivery. Additionally, this property of ionizable lipids can be exploited to help LNPs escape from the endosomes [80,87]. Alameh et al. found that ionizable lipids affect LNP particle size and are a key factor in determining the inherent adjuvant effect of LNPs [88]. Ionizable lipids include single- and multi-charged lipids. Both the mRNA-1273 and BNT162b2 approved COVID-19 mRNA vaccines have single-charged ionizable lipids. Multi-charged lipids have a higher ratio of positively-chargeable polymer amine (nitrogen) groups to negatively-charged nucleic acid phosphate groups, which is more favorable for mRNA encapsulation, cellular uptake, and lysosomal escape. All enhance the immune efficacy of mRNA vaccines. Recently, Chen et al. developed a novel multi-charged lipid termed 4N4T-LNP, which features higher mRNA delivery efficiency and a good safety profile [89]. The above information is summarized as follows (Table 3).

The particle size of LNPs affects their internalization, biodistribution, immunogenicity, degradation, and clearance [90,91]. Controlling the particle size of LNPs also allows for targeted delivery to specific tissues and cells [92]. Typically, the optimal size range for LNPs is 20–200 nm, a range that makes them ideal for permeation and retention and allows them to cross interstitial tissues [90,93].

The application of LNP delivery systems and mRNA technology has greatly increased the development speed of COVID-19 mRNA vaccines. This new class of vaccine has significant advantages in combating the global COVID-19 pandemic. Recent studies have continued to investigate and optimize LNP delivery systems to support the design of subsequent mRNA therapeutic products. Both currently available COVID-19 mRNA vaccines are stored and transported under frozen conditions. The development of mRNA-LNP formulations that do not require frozen storage would reduce production and transportation costs. Therefore, the development of mRNA vaccines should focus on researching key factors affecting the long-term stability of LNP formulations.

### 5.4. Manufacturing Processes

#### 5.4.1. Ethanol

mRNA-LNPs are prepared primarily using microfluidic technology or microjet technology by rapidly mixing the organic phase (primarily ethanol) containing lipids with the aqueous phase containing mRNA. Later, ethanol content can be reduced from the initial 25–1% by dialysis. However, even 1% residual ethanol can affect long-term vaccine stability [35,94], and hydration of ethanol can lead to lipid membrane fusion in the liposome [95]. Because of Fick’s law of diffusion, low-polarity tiny molecules such as ethanol are carried across the lipid bilayer. Ethanol diffusion displaces water molecules on the surface of the liposome and further binds to lipid heads, leading to the formation of an interdigitated structure in the lipid membrane [95]. Since the interdigitated structure has a larger surface area of hydrophobic groups, it is more likely to induce lipid membrane fusion. Moreover, lipid membrane fusion can lead to mRNA leakage [96]. Consequently, reducing residual ethanol from the production process is an important factor affecting mRNA vaccine stability.

#### 5.4.2. pH and Buffer System

pH is also important for mRNA vaccine stability because it affects the rate of mRNA hydrolysis. Under normal conditions, mRNA is more stable in a weakly alkaline environment. The pH of both the approved Moderna and Pfizer/BioNTech COVID-19 RNA vaccines are controlled between 7 and 8. Bauer et al. found that hydrolysis of nucleic acid molecules was significantly accelerated when the pH was decreased from 7.0 to 6.5 [97]. In the presence of Mg^2+^ or Ca^2+^, the phosphodiester bonds of RNA molecules are more susceptible to hydrolytic breakage [31,98].

With respect to the composition of the vaccine buffer system, a suitable buffer system and osmolarity regulator is required because mRNA vaccines may need to be stored below 0 °C and the pH may change after freezing. Pfizer’s COVID-19 vaccine initially used a phosphate buffer containing KCl/NaCl. Subsequently, Tris buffer was used due to the significant pH fluctuations that can occur in phosphate systems after freezing [99]. Tris-HCl stabilizes nucleic acid molecules and scavenges hydroxyl radicals [100,101]. Furthermore, Kolhe et al. used pH probes to demonstrate that the pH of histidine buffer and Tris-HCl buffer fluctuated very little when the temperature was decreased from 0 to −30 °C, with pH increases of 0.5 and 0.6, respectively [102]. In contrast, the pH of a buffer system consisting of sodium phosphate salts fluctuated more during freezing, with a pH decrease of 3.6 when the temperature was decreased from 0 to −30 °C.

#### 5.4.3. Lyophilization

mRNA hydrolysis is the principal factor in mRNA-LNP instability. Structural analysis of mRNA-LNPs has demonstrated that mRNA, ionizable lipids, and water are located in the core of the LNP, with other neutral auxiliary lipids primarily located in the outer wall. The presence of water can initiate hydrolysis reactions in mRNA-LNPs. Lyophilization can be an effective strategy to improve the stability of mRNA-LNP formulations by reducing the water content of the product.

Lyophilization is widely used for live attenuated viral vaccines [103]. During the freeze-drying process, the product structures are exposed to stress. Therefore, cryoprotectant are usually added in order to protect the product from freezing or drying stress and also to increase its stability during storage. The most popular cryoprotectants reported in the literature for freeze-drying nanoparticles include trehalose, sucrose, glucose and mannitol [104]. The level of stabilization afforded by sugars generally relies on their concentrations [105]. Lyophilized vaccines have greater stability than vaccines in conventional liquid formulations [106]. Jones et al. reported that mRNA lyophilized with 10% trehalose as a protectant had high translation and protein expression levels even after storage for 10 months at 4 °C [107]. A lyophilized cytomegalovirus mRNA vaccine (mRNA-1647) developed by Moderna is currently in phase 3 clinical trials. The vaccine is purported to have a shelf life of up to 18 months at 5 °C. Gerhardt et al. found that the lyophilized mRNA vaccine could be stored for over 8 months at room temperature and for over 21 months at 4 °C [108]. The authors described that the physical properties and protein expression levels of the vaccine remained stable. Ball et al. lyophilized LNP-siRNA samples and found that they were stable at −80 °C for up to 11 months; the siRNAs maintained good gene-silencing effects [35]. Muramatsu et al. conducted a similar stability study of lyophilized mRNA-LNP vaccines and found that the samples were stable for 24 weeks at 4 °C or 4 weeks at 42 °C. After that, the vaccine exhibited a significant decrease in mRNA integrity and particle size, as well as a decrease in immunogenicity and mRNA concentration [109]. Recently, a study reported that the physiochemical properties and bioactivities of lyophilized vaccines showed no change at 25 °C over 6 months, and the lyophilized SARS-CoV-2 mRNA vaccines could elicit potent humoral and cellular immunity whether in mice, rabbits, or rhesus macaques [110].

In addition to the COVID-19 mRNA vaccine, the second-generation influenza mRNA vaccine developed by Sanofi incorporates lyophilization. However, lyophilization requires re-dissolution before administration, during which particle aggregation might occur, which is detrimental to vaccine efficacy. In addition, lyophilization is expensive, laborious, and time-consuming.

### 5.5. Temperature and Physical Shock

Temperature is an important environmental factor affecting vaccine stability. Packer et al. studied the effect of different ambient temperatures (−20, 5, 25, and 40 °C) on the stability of mRNA vaccines [39]. RP-IP HPLC assays revealed some degree of mRNA degradation in all groups, with a progressively higher percentage of mRNA integrity loss with increasing ambient temperature. Of the two approved COVID-19 mRNA vaccines, data showed that the Moderna vaccine is stable at −20 °C for 6 months and the Pfizer-BioNTech vaccine is stable at −80–−60 °C for 6 months. However, when stored at 2–8 °C, the shelf life of the Moderna and Pfizer-BioNTech vaccines is rapidly reduced to 30 days and 5 days, respectively. The Onpattro siRNA drug formulation has a shelf life of 3 years at 2–8 °C [111], suggesting that a key factor affecting mRNA vaccine stability may be mRNA structure itself. Zhang et al. developed a novel liquid formulation of a COVID-19 mRNA vaccine based on LNP-encapsulated mRNA that encodes the RBD sequence of SARS-CoV-2 [112]. The vaccine can be stored for 7 days at room temperature and for one month at 2–8 °C with no effect on vaccine immunogenicity. Furthermore, mRNA purity, encapsulation efficiency, polymer dispersity index, and in vivo antibody levels did not change significantly when stored for 6 months at 2–8 °C [113]. Strict control of temperature as well as rational process design are critical for the temperature sensitivity of mRNA vaccines and its improved stability.

In addition to temperature, physical shock is another important environmental factor affecting vaccine stability. Grau et al. investigated the stability of the Pfizer-BioNTech and Moderna COVID-19 mRNA vaccines in a continuous movement environment [114]. This environment had no significant effect on mRNA vaccines. When the intensity of shaking was increased up to 180 min, changes in mRNA integrity and degraded RNA fragments were observed. Kudsiova et al. performed a similar study for the BNT162b2 vaccine and found that vigorous shaking, vibration, and repeated syringe draw-and-release resulted in increased BNT162b2 lipid nanoparticles particle size and changes in polymer dispersity index, accompanied by the release of a large amount of mRNA from the lipid nanoparticles and a loss of free mRNA, which inevitably affects the protective effect of the vaccine [115].

## 6. Conclusions

The COVID-19 pandemic has accelerated the development and application of mRNA vaccines. Owing to its huge production capacity and fast speed of research and development, the mRNA vaccine has great potential in the prevention and control of infectious diseases. However, due to the inherent instability of RNA, the mRNA vaccine usually needs harsh production conditions, and lower storage temperatures and transportation conditions, which shortens the shelf life of the vaccine and restricts the global accessibility of vaccines. Therefore, improving the stability of the mRNA vaccine is a challenge in this field.

The degradation pathways of mRNA include physical degradation and chemical degradation. Hydrolysis is the main form of mRNA destruction. Both self-cleaving nuclease and protein nuclease can destroy the stability of RNA through hydrolysis. Research on pH, temperature, ion concentration and other factors is of great significance to improve stability over the full life span of mRNA vaccines, including formulation, preparation, storage and in vivo immune effect. In terms of vaccine design and manufacturing techniques, the length and structure of mRNA, codon optimization, nucleoside modification, GC content, LNP formulation, buffer system and other factors jointly affect the stability of mRNA vaccines.

A suitable LNPs delivery system is vital for the uptake and expression of mRNA in vivo and its in vitro stability. The level of excipients and manufacturing techniques also affect the stability of mRNA vaccines. Controlling the quality of excipients and developing advanced manufacturing techniques will contribute to improve the long-term stability of such vaccines. At present, developing advanced drying technology or adopting low-temperature storage is the main means to deal with this problem. Furthermore, exploring the degradation model of the vaccine and the action mechanism between lipid and mRNA will help improve the stability of mRNA vaccine research.

Recently, McMahon et al. proposed a risk-based stability research idea, suggesting that the accelerated stability model, empirical data, kinetic analysis, and other risk-based ideas could be used to predict the stability of the mRNA vaccine, and to obtain the quality attributes related to the products’ stability and the expiration limit attributes, which would provide guidance for the research and evaluation of the stability of mRNA vaccines [3].

In conclusion, we should deeply understand the underlying degradation mechanism and influencing factors of mRNA vaccines, improve the stability through structural and technological optimization, and carry out research on stability characteristics to lay the foundation for the safety and effectiveness of such vaccines.

## Figures and Tables

**Table 1 viruses-15-00668-t001:** Regulations and guidelines related to vaccine stability.

Issue Date	Name of Regulation	Issuing Agency /Country	Reference
1989	Stability of vaccines (WHO/EPI/GEN/89.08)	WHO	[4]
1995	Stability testing of biotechnological/biological products—Scientific guideline (ICH Q5C)	ICH	[6]
1998	Thermostability of vaccines (WHO/GPV/98.07)	WHO	[5]
2003	Evaluation for Stability Data Q1E	ICH	[16]
2003	Stability Testing of New Drug Substances and Products Q1A(R2)	ICH	[17]
2011	Guidelines on Stability Evaluation of Vaccines	WHO	[8]
2012	Controlled Temperature Chain (CTC) Guidelines	WHO	[18]
2013	Extended Controlled Temperature Conditions (ECTC) Guidelines	WHO	[19]
2015	Technical Guidelines for Stability Studies of Biological Products	CDE/China	[9]
2018	Liposomal Drug Products: Chemistry, Manufacturing, and Controls; Human Pharmacokinetic and Bioavailability; and Labeling Documentation	FDA/USA	[14]
2020	2020 Chinese Pharmacopoeia Section 4 Regulation 9402 Guidelines for stability testing of biological products	Chinese Pharmacopoeia Commission /China	[10]
2021	Evaluation of the quality, safety, and efficacy of messenger RNA vaccines for the prevention of infectious diseases: regulatory considerations	WHO	[13]
2022	Technical Guidelines for the Pharmacological Study and Evaluation of In Vivo Gene Therapy Products (draft edition)	CDE/China	[15]

**Table 2 viruses-15-00668-t002:** Clinical trials of COVID-19 mRNA vaccines *.

Name	Developer	Country	Phase
mRNA-1273	Moderna	USA	IV
BNT162b2	Pfizer/BioNTech	USA/Germany	IV
mRNA-1273.351	Moderna	USA	IV
CVnCoV	CureVac AG	Germany	III
PTX-COVID19-B	Providence Therapeutics	Canada	III
SW-BIC-213	Shanghai East Hospital and Stemirna Therapeutics	China	III
ARCoV	Academy of Military Science (AMS) and Suzhou Abogen	China	III
ARCT-154	Arcturus Therapeutics	USA	III
LVRNA009	AIM Vaccine and Liverna Therapeutics	China	III
mRNA-1273.214	Moderna	USA	III
DS-5670a	Daiichi Sankyo Co., Ltd.	Japan	II/III
HDT-301	SENAI CIMATEC	Brazil	II/III
mRNA-1273.211	Moderna	USA	II/III
mRNA-1273.529	Moderna	USA	II/III
mRNA GEMCOVAC-19	Gennova Biopharmaceuticals Limited	India	II/III
ChulaCov19 mRNA vaccine	Chulalongkorn University	Thailand	II
MRT5500	Sanofi Pasteur	France	II
ARCT-021	Arcturus Therapeutics	USA	II
SYS6006	CSPC ZhongQi Pharmaceutical Technology Co., Ltd.	China	II
GLB-COV2-043	GreenLight Biosciences, Inc.	USA	I/II
EXG-5003	Elixirgen Therapeutics, Inc	USA	I/II
ARCT-165	Arcturus Therapeutics	USA	I/II
ARCT-021	Arcturus Therapeutics	USA	I/II
EG-COVID vaccine	EyeGene Inc.	South Korea	I/II
AAHI-SC2 and AAHI-SC3	ImmunityBio, Inc.	USA	I/II
mRNA-1073	Moderna	USA	I/II
CoV2 SAM(LNP) vaccine	GlaxoSmithKline	UK	I
LNP-nCoV saRNA	Imperial College London	UK	I
mRNA-1283	Moderna	USA	I
LNP-nCOV saRNA-02	MRC/UVRI and LSHTM Uganda Research Unit	UK	I
HDT-301 vaccine	HDT Bio	USA	I
VLPCOV-01	VLP Therapeutics Japan GK	USA	I
CV2CoV	CureVac AG	Germany	I
MIPSCo-mRNA-RBD-1	University of Melbourne	Australia	I
Lyophilized Vaccine	Jiangsu Rec-Biotechnology Co., Ltd.	China	I
Lyophilized Vaccine	Wuhan Recogen Biotechnology Co., Ltd.	China	I
RQ3013	Walvax Biotechnology; Shanghai RNACure Biopharma	China	I
RVM-V001	RVAC Medicines	Singapore	I
ABO1009-DP	Suzhou Abogen Biosciences Co., Ltd.	China	I
Self-Amplifying Vaccines	Gritstone bio, Inc.	USA	I
CV0501	GlaxoSmithKline	USA	I

* Data source from World Health Organization. COVID-19 Vaccine Tracker and Landscape [Eb/Ol] (7 February 2023). “https://www.who.int/publications/m/item/draft-landscape-of-covid-19-candidate-vaccines (accessed on 8 February 2023)”.

**Table 3 viruses-15-00668-t003:** Composition and function of LNP in mRNA vaccines *.

Component	Proportion (%)	Function	Modifications and Related-Functions	Reference
phospholipid	10–20	structural lipids	3-phosphate group modification: improve transfection efficiency; increase the distribution in vivo	[78,79]
cholesterol	20–50	structural lipids	C-24 alkyl derivatives: regulate the integrity of the membrane; impact on delivery efficiency and biodistribution	[80,81,82]
PEG-lipid	~1.5	Prevent aggregation; stabilization	Covalent Coupling modification: improving the half-life of LNPs	[83,84,85,86]
Ionizable lipid	30–50	Encapsulation; delivery	Introduction of multiple tertiary amine nitrogen atoms: higher delivery efficiency and safety	[80,87,89]

* Data source from pre-clinical stage study.

## Data Availability

Not applicable.

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
