# Peer review of "Research Advances on the Stability of mRNA Vaccines"

_viruses, 2023, doi:10.3390/v15030668_

Round 1

Reviewer 1 Report

The manuscript by Cheng et al. "Research advances on the stability of mRNA vaccines" is a well-detailed work that examines all key elements for stability of mRNA vaccines. Such work is largely required given the importance of mRNA lipid nanoparticles for vaccines and therapeutics, therefore this review could be of interest to a large community. The manuscript has some weaknesses and flaws to address.

The major issues is that sometimes in the text the terms “liposomes” is used wrongly, it should be more properly talked about lipid nanoparticles. It is common for liposomes to be called lipid nanoparticles, but strictly speaking, this is not correct. Liposomes and lipid nanoparticles are two different types of lipid-based delivery, and while there is some overlap, they have different structures and stability.

Please correct in the abstract and in section 2 “Regulatory documents” you need to explain the relevance (in line 88) why you refers to Liposomal drug products. I fear that the two terms liposomes and lipid nanoparticles have been sometimes (erroneously) used as synonymous.

In the introduction part, line 36, please explain what the authors exactly means for stock solution and final products. The reader might be confused. Please provide more details.

In line 41 the author state that both approved vaccines are transported under ultra-low temperature. The one made by Pfizer needs to be kept extremely cold: minus 70 degrees Celsius, Moderna has said that its vaccine needs to be frozen too, but only at minus 20 Celsius. Please modify the text.

In general, the introduction part is not at the quality level of the rest of the manuscript and can be improved.

In section 4.2, line 18 “(DSPC) and ionisable lipid”. There should be a mistake in “and”, please verify.

In line 182 interaction with DNA bases is indicated. Could you provide references also for RNA as the work is on m-RNA? Or detailed why it is relevant the reference with DNA bases to all readers.

Spacing needs sometimes to be adjustes e.g see lines 140-145.

Author Response

Dear editor and reviewer1,

We are very grateful to editor and reviewers for those valuable advices and comments. We have adopted the suggestions and revised them. The revised parts were marked in red in the revised manuscript.

Reviewer 2 Report

Comments re Cheng et al., 2023 Research advances on the stability of mRNA vaccines

This is a review article on the stability of mRNA-vaccine, in particular the LNP-mRNA products. The first part covers regulatory guidance documents for this family of products. This is a timely compilation of material relevant for those who wish to receive approval for mrNA-LNP products in the future.

Sections 3-5 of the manuscript cover aspects such as degradation reactions and strategies to increase stability. These sections are covered earlier by others in review articles and hardly bring new views to the table.

In general, the article is well-written. I have some comments re textual details.

42                  have ... has....

68                  physical shakes...... physical stress

88                  I would suggest not to use the term ‘liposomes’ in the context of LNP. The definition of liposome is: a water- filled vesicular structure based on (phospho)lipid bilayers...... LNP -although containing (phospho)lipids- are differently organized.

136                TABLE 2: Source of this information?

176/179         May I remind the authors that Patisaran has a shelf life of 3 years. Apparently, these degradation reactions are not the main issue when discussing mRNA-LNP stability.  

198                Please provide reference details for the EMA document.

213                ... significant shorten    ...... significantly shorten......

290                 ..... cap analogs, ribonucleoside triphosphates 290 (NTPs), modified nucleotides (m1ψTP), and polynucleotide tails.... These are not excipients. These are part of the API, active pharmaceutical ingredient.

313                    is sucrose an anti-oxidant? .....maybe, but definitely not the first choice as antioxidant.

318                      ...... LNPs is...... .....LNPs are....

327                      DOPE is not a ‘novel’ phospholipid.

339                    ionizable lipids are only neutral at physiological pH when present in the core of the LNPs. They are tertiary amine which have pKa-s around 9-10 when ‘solo’.

355                    Table 3: it may be useful to indicate what the stage of development (clinical, preclinical) of these modified products is. In addition, ...... ‘çovalent coupling modification’....... are these new PEG-lipids only modified with respect to the coupling reactions?

359                     ... robust enough..... what is the relationship between particle size and robustness?

377-380             The relationship between disruption of the membrane and Fick's law is unclear to me. In addition, the literature mentioned (95) concerns liposomal structures and not LNP.

392                    Reference 98: the title refers to DNA. Is this relevant for mRNA in LNP?

416                      Mannitol is not a sugar.

435-439               Please, provide lit.  reference.

481                       .... whole life process..........stability over the full life span......

Author Response

Dear editor and reviewer2,

We are very grateful to editor and reviewers for those valuable advices and comments. We have adopted the suggestions and revised them. The revised parts were marked in red in the revised manuscript.
